# The impacts of hydropower on freshwater macroinvertebrate richness: A global meta-analysis

Gabrielle Trottier[1]*, Katrine Turgeon[2], Daniel Boisclair[3], Cécile Bulle[4], Manuele Margni[1,5]

1 CIRAIG, Département de mathématique et génie industriel, Polytechnique Montréal, Montréal, Québec, Canada, 2 ISFORT, Université du Québec en Outaouais, Ripon, Québec, Canada, 3 Département de sciences biologiques, Université de Montréal, Montréal, Québec, Canada, 4 CIRAIG, Département de stratégie, responsabilité sociale et environnementale, École des sciences de la gestion, Université du Québec à Montréal, Montréal, Québec, Canada, 5 Institute of Sustainable Energy, HES-SO Valais-Wallis, Sion, Switzerland

* gabrielle.trottier@polymtl.ca

**Data Availability Statement:** All relevant data are within the manuscript and its Supporting Information files.

**Funding:** MM and GT were funded by the Natural Sciences and Engineering Research Council of

## Abstract

Hydroelectric dams and their reservoirs have been suggested to affect freshwater biodiversity. Nevertheless, studies investigating the consequences of hydroelectric dams and reservoirs on macroinvertebrate richness have reached opposite conclusions. We performed a meta-analysis devised to elucidate the effects of hydropower, dams and reservoirs on macroinvertebrate richness while accounting for the potential role played by moderators such as biomes, impact types, study designs, sampling seasons and gears. We used a random/mixed-effects model, combined with robust variance estimation, to conduct the meta-analysis on 107 pairs of observations (*i.e.*, impacted versus reference) extracted from 24 studies (more than one observation per study). Hydropower, dams and reservoirs did significantly impact (P = 0.04) macroinvertebrate richness in a clear, directional and statistically significant way, where macroinvertebrate richness in hydropower, dams and reservoirs impacted environments were significantly lower than in unimpacted environments. We also observed a large range of effect sizes, from very negative to very positive impacts of hydropower. We tried to account for the large variability in effect sizes using moderators, but none of the moderators included in the meta-analysis had statistically significant effects. This suggests that some other moderators (unavailable for the 24 studies) might be important (*e.g.*, temperature, granulometry, wave disturbance and macrophytes) and that macroinvertebrate richness may be driven by local, smaller scale processes. As new studies become available, it would be interesting to keep enriching this meta-analysis, as well as collecting local habitat variables, to see if we could statistically strengthen and deepen the conclusions of this meta-analysis.

Canada (NSERCC). GT was also funded by the Fonds Quebecois de la Recherche sur la Nature et les Technologies (FQRNT), Fondation Polytechnique and Hydro-Quebec, as well as the Institut de l'Environnement, le Developpement Durable et l'economie Circulaire (EDDEC) and Banque TD. The funders had no role in study design, data collection and analysis, decision to publish, or preparation of the manuscripts.

**Competing interests:** The authors have declared that no competing interests exist.

## Introduction

Freshwater ecosystems are vital resources for humans and support a biota that is rich, sensitive and characterized by a high level of endemicity [1]. Ecosystems functions and integrity often depend on biodiversity, which can be described by three indices: species richness (*i.e.*, number of species), community assemblage (*i.e.*, proportions of different species or taxonomic groups in the community) and functional diversity (*i.e.*, variability in organisms' traits that can influence ecosystem functioning; [1]). For millennia, humans have used freshwater ecosystems, through water extraction for drinking and irrigation purposes, water regulation for hydropower production, flood control and recreation [1], but these usages often come with a cost on freshwater ecosystems biodiversity [2, 3].

Hydroelectric dams and the creation of reservoirs, at all stages (*i.e.*, from the construction, to operation and decommission of a dam), can affect freshwater biodiversity [4]. Dams create a physical barrier, which can impair the natural flow of water, sediments and nutrients [5, 6] and limit the movement of organisms [7]. The alteration of the natural hydrological regime can affect freshwater biodiversity through various biological mechanisms (*e.g.*, mortality through desiccation, mismatch timing in life history strategies, lotic to lentic community changes, reduction/extirpation of endemic and specialist species; [8, 9]) and through degraded water quality (anoxic or hypoxic releases [dissolved oxygen], hypolimnetic or epilimnetic releases [temperature], pH, organic carbon, turbidity; [4, 10–13]).

Studies investigating the impact of hydropower on the richness of macroinvertebrates drew contrasting conclusions. Some studies reported that richness is negatively impacted by hydropower, through general flow regulation [14–16] and water level fluctuation (or drawdown; [17–22]). Others observed higher richness downstream of a dam [23, 24] or in regulated rivers (as opposed to natural ones; [25]). Finally, Marchetti et al. [26] found little difference between impacted flows (*i.e.*, dam-induced permanent low flow) and "natural-like" flows (*i.e.*, high flows in winter and spring, low flows in summer and falls). A meta-analysis could elucidate patterns and interactions that may exist between hydropower, macroinvertebrate richness, and the context in which studies have been conducted.

Many challenges can be encountered when conducting a meta-analysis, and many ecological facets of studied ecosystems can influence the magnitude and significance of human impacts [27], along with study-specific methodological characteristics (*e.g.*, different study design). The influence and the variability brought about by these facets and characteristics can be accounted for through variables, also called moderators in meta-analysis [28]. For instance, the location of each study site can influence the observed effects [27, 29, 30]. A latitudinal biodiversity gradient is a good example of the influence of spatial location. Species richness is known to be highest in the tropics and lowest at the poles [30, 31]. Thus, losing few species to hydropower activities will not have the same repercussions on low diversity macroinvertebrate communities than higher diversity communities (*e.g.*, losing 2/10 species [20% loss] versus 2/25 species [8% loss]). Hydropower can lead to different types of impacts. A study can analyze the impacts of hydropower upstream of a dam, that is in the reservoir, or downstream of a dam. Impacts also varies depending on the type of water management in place, storage reservoir with winter water level drawdown, hydropeaking or typical run-of-river hydrological regime. These variations in studies, along with the location under study, can introduce variability and heterogeneity in the results, which can be accounted for through moderators. The experimental design can also influence how human-induced impacts magnitude are reflected in a study [27]. As demonstrated in Christie et al. [32], different sampling designs may affect the conclusion of a study. Using simulations, they demonstrated that Before-After (BA), Control-Impact (CI; analogous to space-for-time substitution) and After designs are far less

accurate than Randomized Controlled Trials (RCT) and Before-After Control-Impact (BACI) designs. RCT and BACI are much harder to implement in ecology because true randomization can be difficult with larger scale designs and getting data before the impacts or human intervention is sometimes impossible. Thus, we must account for the effect of the experimental design on a study outcome, especially in a meta-analysis, where the effects of multiple different studies are combined. Sampling season can also influence the results across studies, as macroinvertebrate communities differs in terms of abundance and diversity throughout the year (*i.e.*, maximum diversity in late summer and autumn versus underrepresented diversity in spring and early/mid-summer; [33]). At a more local scale, the habitat stratum that is sampled is also most likely to influence the results [27], especially when studying macroinvertebrates. These organisms possess characteristics that make them highly adapted to their environment [34] and because lakes, reservoirs and river beds are so heterogeneous, macroinvertebrates are often patchily distributed, requiring extensive sampling [35]. Thus, the type of sampling gear used to sample will likely affect the type of organisms inventoried in each study. However, using sampling gear as a proxy for habitat stratum might need a caveat as many biomonitoring protocols (*e.g.*, United State Environmental Protection Agency) use nets for multiple-habitat assessments. In this study, we assumed that different sampling methods would collect different macroinvertebrate communities.

The objective of this manuscript is to conduct a meta-analysis about the impacts of hydropower dams and their reservoirs on the richness of macroinvertebrates while accounting for a series of moderators defining the context of the studies included. The moderators included in this manuscript are the following: 1) biomes (*i.e.*, boreal, temperate, and tropical), a proxy for location/latitudinal gradient, 2) type of impact, which is reflected by the position of a sample in relation to the dam (*i.e.*, upstream or downstream of the dam). Downstream stations are impacted by a reduced flow and hydropeaking dynamics, whereas upstream stations are impacted by drawdown and water level fluctuations due to reservoir management, 3) type of study designs such as cross-sectional (*i.e.*, reference natural lake versus impacted reservoir) and longitudinal spatial gradient (*i.e.*, upstream of a dam [reservoir] versus downstream of a dam [river]), which are two different variants of CI study design, 4) sampling seasons (*i.e.*, spring, summer, fall, winter; we were interested in the coarser temporal effect of seasons rather than daily changes in colonization following water regulation) and 5) sampling gears, a proxy for habitat stratum (*i.e.*, grabs and nets). Even if richness constitutes only one of three components of biodiversity, we chose to focus on macroinvertebrate richness because it is easier to quantify and extract from the scientific literature than community composition and also a lot more common than functional diversity.

## Methodology

Achieving our objective requires to first establish a research strategy, second to do a data collection comprising information regarding the richness of macroinvertebrates in hydropower impacted habitat versus reference ones, biome, the type of impact, study design, sampling seasons and gear. Third, it requires to compute effect sizes for each study and finally, combine them to assess if the mean effect size is significantly different from zero [36] and if the presence of other moderators can influence these results.

### Research strategy

In this study, we used the PRISMA (Preferred Reporting Items for Systematic and Meta-Analyses) methodology, flow diagram (Fig 1) and checklist (S1 Table) proposed by Moher et al. [37], to report systematic literature reviews and meta-analyses. A systematic literature review

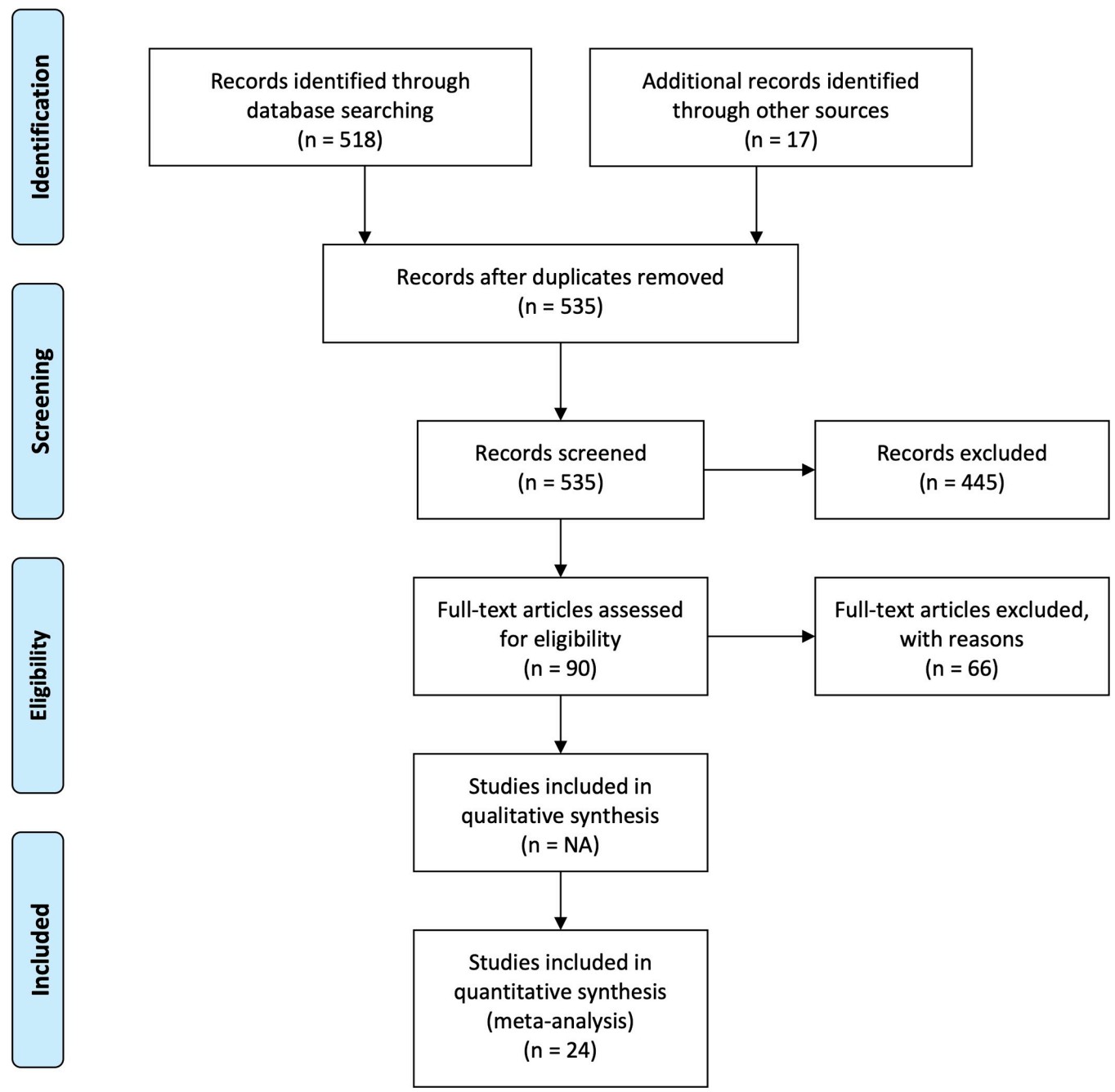

**Fig 1. PRISMA diagram.** Specific to this meta-analysis, from Moher et al. [37].

was conducted using the Web of Science Core Collection database, which includes all journals indexed in Science Citation Index Expanded (SCI-EXPANDED), Social Sciences Citation Index (SSCI), Arts & Humanities Citation Index (A&HCI), Conference Proceedings Citation Index–Science (CPCI-S), Conference Proceedings Citation Index–Social Science & Humanities (CPCI-SSH) and Emerging Sources Citation Index (ESCI; [38]). The research strategy was constrained between 1989 (earliest searchable year) and 2021 and contained a combination of the four following field of research: 1) hydropower (hydropower OR hydroelectric* OR dam

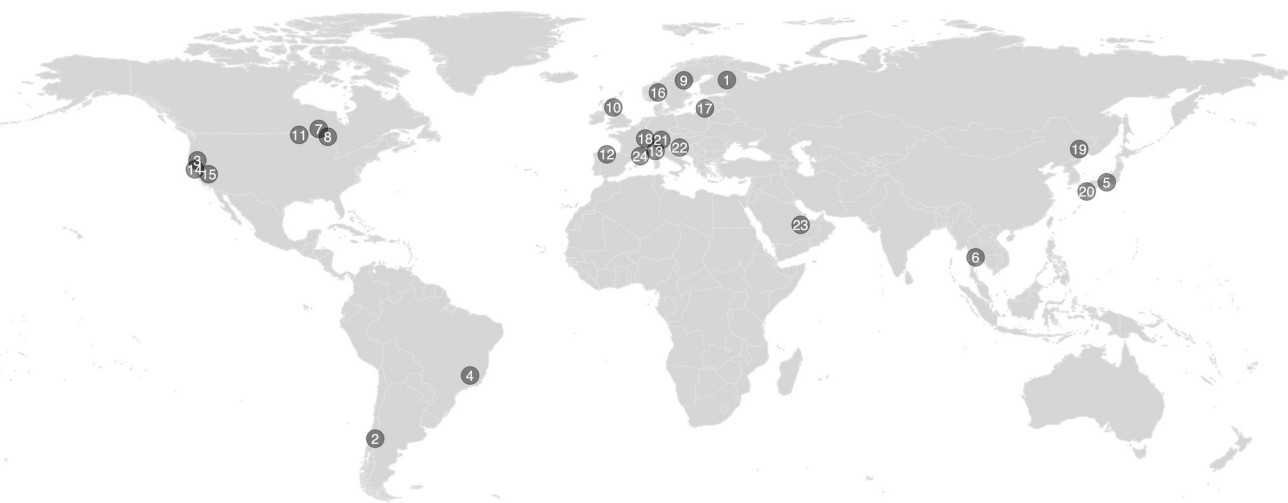

**Fig 2. World map showing the geographical disposition of the studies used in this meta-analysis.** (1) Aroviita and Hämäläinen (2008) [39], (2) Valdovinos et al. (2007) [20], (3) Marchetti et al. (2011) [26], (4) Molozzi et al. (2013) [40], (5) Takao et al. (2008) [15], (6) Kullasoot et al. (2017) [16], (7) White et al. (2011) [22], (8) Smokorowski et al. (2011) [25], (9) Englund and Malmqvist (1996) [18], (10) Jackson et al. (2007) [14], (11) Kraft (1988) [17], (12) Mellado-Diaz et al. (2019) [41], (13) Bruno et al. (2019) [42], (14) Milner et al. (2019) [43], (15) Steel et al. (2018) [44], (16) Schneider and Petrin (2017) [45], (17) Vaikasas et al. (2013) [46], (18) Doledec et al. (2021) [47], (19) Wang et al. (2016) [48], (20) Nukazawa et al. (2020) [49], (21) Quadroni et al. (2020) [50], (22) Vilenica et al. (2020) [51], (23) Uieda and Marcal (2020) [52] and (24) Cazaubon and Giudicelli (1999) [53].

OR dams OR reservoir* OR impound* OR run-of-river OR "run of river" OR drawdown* OR hydropeak* OR dam* OR "water level fluctuation" OR "water-level fluctuation" OR "water level variation" OR "water-level variation" OR "water level regulation" OR "water-level regulation" OR "water level manipulation" OR "water-level manipulation" OR "water management"), 2) biodiversity (biodiversity OR richness), 3) freshwater ecosystems (freshwater OR aquatic) and 4) aquatic insects (*invertebrate* OR *benth* OR insect* OR arthropod*). Research strategy also excluded all studies pertaining to beaver and agricultural dams (NOT beaver* NOT agricult*). The research strategy resulted in 518 research articles, as per September 1st 2021.

The results were extracted as a list to further evaluate the relevance of every study based on a list of criteria. The following criteria were applied to assess the inclusion of any study in the meta-analysis: 1) the study had to refer specifically to hydropower related impacts (*i.e.*, reservoir, run-of-the-river or hydropeaking, multi-purpose reservoirs were also checked for hydropower impacts), 2) the scope of the study had to address freshwater ecosystems and macroinvertebrates and 3) the study had to be empirical (*i.e.*, excluding literature reviews, modelling exercises) and provide an explicit richness, error and sample size value for both a reference and impacted site (*i.e.*, cross-sectional studies [reference versus impacted]) or gradient of impact (longitudinal spatial gradient studies [upstream of the dam/reservoir versus one or multiple sites downstream of the dam]). Out of the 518 research studies that resulted from the research strategy, only 24 met the above criteria (see geographical disposition of studies in Fig 2).

## Data collection

Richness metrics and moderators were collected for each of these 24 studies by the corresponding author (S2 and S3 Tables). We extracted richness (*i.e.*, number of taxa), error measure (*e.g.*, standard deviation), sample size (*i.e.*, number of reference observations and impacted observations used to compute richness and its associated error) and a suite of

moderators such as biome (*i.e.*, boreal, temperate and tropical; [54]), type of impact (*i.e.*, water level fluctuations upstream, due to reservoir management, or flow regulation downstream due to dam operations), type of study, (*i.e.*, cross-sectional [reference natural lake versus impacted reservoir] or longitudinal spatial gradient [reference; upstream of the dam/reservoir versus impacted; downstream of the dam]), sampling season (*i.e.*, spring [March to May], summer [June to August], fall [September to November] and winter [December to February], according to the hemisphere) and sampling gear (*i.e.*, net, grab or colonization basket). These moderators were chosen based on the availability of said moderators in each of the 24 studies, their potential influence on macroinvertebrate richness and complemented with expert judgement. In studies where meaningful data were presented exclusively in graphical format, values were extracted using Engauge Digitizer 10.4 [55].

**Effect size.** To compute effect sizes, we calculated the standardized mean differences, also called Cohen's *d*–which expresses the distance between two means (*i.e.*, impact and reference) in terms of their common standard deviation [56]. For most studies–except Takao et al. [15], Schneider and Petrin [45] and Vilenica et al. [51], we computed at least two effect sizes per study, leading to a total of 107 effect sizes, with a certain level of within-study dependency (*i.e.*, effect sizes in one study are not entirely independent from each other). Cohen's *d* is calculated as per Eq 1 [56]:

$$d = \frac{\bar{X}_1 - \bar{X}_2}{\sqrt{\left(\frac{(n_1-1)s_1^2 + (n_2-1)s_2^2}{n_1 + n_2 - 2}\right)}} \tag{1}$$

where $\bar{X}_1$ and $\bar{X}_2$ are the mean richness, $s_1^2$ and $s_2^2$ are the standard deviation (SD) and, $n_1$ and $n_2$ are the number of observations used to compute the mean and SD for impacted and reference samples, respectively. The common variance ($V_d$) associated with the effect size (*d*) is calculated using Eq 2:

$$V_d = \frac{n_1 + n_2}{n_1 \cdot n_1} + \frac{d^2}{2(n_1 + n_2)} \tag{2}$$

In the case of smaller sample size (usually < 20 studies), a correction factor is applied to Cohen's *d* to reduce the positive bias (negligible with bigger sample size) and to provide a better estimate. The corrected effect size is then called a Hedges' *g* [57]. A small sample correction factor (*J*) was computed using Eq 3:

$$J = 1 - \frac{3}{4df - 1} \tag{3}$$

where *df* refers to the degrees of freedom ($n_1 + n_2 - 2$). Thus, the corrected effect size *g* and variance $V_g$ are calculated following Eqs 4 and 5, respectively:

$$g = J \cdot d \tag{4}$$

$$V_g = J^2 \cdot V_d \tag{5}$$

A positive *g* means the impacted environment or sample has more richness in comparison to a reference environment or sample. The *metafor* package [28] was used to compute the effect sizes and sampling variances (*i.e.*, *escalc* function) and the *ggplot2* package [58] was used to graphically visualize the results of the meta-analysis.

## Data analysis

**Publication bias.**    Studies with large significant results are more likely to be published than studies with non-significant results, this is called publication bias [36]. A funnel plot was used to evaluate the presence of a publication bias in the meta-analysis [59], and a regression test for funnel plot was used to detect potential asymmetry (*i.e.*, if only large significant studies, range of outcomes is not well represented and studies with non-significant results might not even be present; [60]). The *metafor* package [28] was used for asymmetry analysis (*i.e.*, *funnel* and *regtest* functions).

**Heterogeneity.**    There are two sources of heterogeneity in a meta-analysis 1) the heterogeneity due to sampling error, or the within-study heterogeneity (*i.e.*, methodological heterogeneity), which is always present in meta-analyses as every study uses different samples, and 2) the true heterogeneity due to specific study characteristics (*e.g.*, biome) and dissimilarities in methodologies among studies (*e.g.*, study design), which can introduce variability among true effect sizes [28, 61]. The combination of the methodological and the true heterogeneity is referred to as the total heterogeneity. It is interesting to examine this total heterogeneity and identify the different moderators and their relative contributions to the magnitude and direction of these effect sizes [36]. We evaluated the statistical significance and the magnitude of the total heterogeneity of effect sizes among studies (*i.e.*, heterogeneity analysis) using the $Q$ statistic, followed by the $I^2$ index to identify how much of this total heterogeneity is due to true heterogeneity [61]. The *metafor* package [28] was used for heterogeneity analysis (*i.e.*, *rma* function).

**Random/Mixed-effects model: Dealing with heterogeneity.**    Two types of meta-analytic models can be used in a meta-analysis, fixed-effects or random/mixed-effects. A fixed-effects model considers only the studies included in the meta-analysis and within study sampling variability, not between studies [36]. No inference can be made outside this set of studies (*i.e.*, conditional inferences; [28]). A random/mixed-effects model considers the set of analyzed studies as a sample of a larger population of studies [28]. Thus, it allows the researcher to make inferences regarding what would be found if an entire new meta-analysis, with a different set of studies, was performed (*i.e.*, unconditional inferences; [28]). It also allows two sources of variation, within and among studies [36]. Such an approach is especially appropriate when dealing with heterogeneity among studies [61] and, with a random/mixed-effects model approach, it is also possible to include moderators, which can account for some of that heterogeneity [28]. It is important to highlight that these moderators do not impact species richness, they rather explain additional heterogeneity in the meta-analysis effect sizes. Here, a random/mixed-effects modelling is preferred to a fixed-effects modelling approach since 1) a significant amount of heterogeneity was found in the previous heterogeneity analysis and 2) because it offers the possibility to model and explain some of that heterogeneity using moderators [56].

**Robust variance estimation: Dealing with dependency.**    If our effect sizes were all independent from each other, we could have simply used a random/mixed-effects model. However, because this meta-analysis is dealing with multiple effect sizes per study, where observations are not methodologically and spatially independent from each other, it is inappropriate to use a regular meta-analysis approach (*i.e.*, random/mixed-effects model), where the effect sizes are assumed to be independent. One way to account for dependency of effect sizes is to combine the random/mixed-effects model with the robust variance estimation (RVE) method. The RVE estimates the overall effect size over studies using a weighted mean of the observed effect sizes [62]. It doesn't require knowledge about the within-study covariance, it can be applied to any type of dependency and effect sizes, it simultaneously accommodates for multiple sources of dependencies, it does not require the effect sizes to comply to any particular distribution

assumptions, it leads to unbiased fixed-effects and standard errors estimates and can also give an estimate of among-study variance [62, 63]. Because the most common source of dependence within the effect sizes in this meta-analysis is the correlated nature of the observations (*i. e.*, multiple measures within a study; methodological and spatial correlation) and not the hierarchical nature (*i.e.*, common nesting structure between studies; a sample within a transect, within site and within a lake), a correlated effects weighting method was used [63]. Thus, we used a RVE based on a correlated effects model and adjusted for small sample size ($< 40$ studies; [63]). Finally, a sensitivity analysis was computed to assess the effect of a varying rho ($\rho$) value, which is a user-specified value of the within-study effect sizes correlation (*i.e.*, the correlation between two samples taken in the same water body in one specific study–spatial and methodological correlation; [63]). The *robumeta* package [63] was used to fit the RVE meta-regression model (*i.e.*, *robu* function) and to compute the sensitivity analysis (*i.e.*, *sensitivity* function). All statistical analyses were made using R version 3.0.2 [64].

## Results

### Methodological results

The purpose of this first set of results is to validate our methodological approach and choices, they will not be the subject of discussion. No statistical asymmetry was observed in the funnel plot ($z = -1.74$; $P = 0.08$; S1 Fig), a wide range of results and significance levels were represented by the studies included in this meta-analysis. The total heterogeneity among the effect sizes was statistically significant ($Q_{df = 23} = 140.16$; $P < 0.0001$), which indicated greater total heterogeneity than expected by the sampling error alone. The estimated amount of this total heterogeneity among the effect sizes was $T^2 = 2.01$; 95% confidence interval [CI] = 1.05–4.90. Of that total heterogeneity, a large amount ($I^2 = 89.44\%$; CI = 81.59–95.39) was due to true heterogeneity between the studies, rather than just methodological heterogeneity. Thus, further examination of the true heterogeneity is warranted and was done through the analysis of multiple moderators.

### Meta-analysis results

The meta-analysis of 24 studies (107 pairs of observations; S1 Table) suggests that hydropower dams and reservoirs did have a statistically significant effect on macroinvertebrate richness. The mean effect size (*i.e.*, Hedge's *g*) estimate of our RVE model was -0.84 (95% CI = -1.62 to -0.05; $P = 0.04$), without accounting for the different moderators (Fig 3). The large confidence interval not overlapping zero indicates that the mean effect size is statistically significant, but also highlights a wide range of effect sizes across studies. The sensitivity analysis shows that the estimates of the mean effect size and standard errors, as well as the estimate of between-study variance in study-average effect sizes ($\tau^2$), are relatively insensitive to different value of $\rho$ (S4 Table).

Moderators had very little influence in the effects of dams and reservoirs on macroinvertebrate richness. Biome did not significantly explain variability effects sizes. For this moderator, it was only possible to make statistical interpretation for the temperate level (estimate = -0.23; 95% CI = -1.97 to 1.51; $P = 0.74$; Fig 4A) and tropical level (estimate = 0.02; 95% CI = -4.53 to 4.57; $P = 0.99$; Fig 4A). We can interpret with statistical confidence that temperate and tropical biome did not significantly differ from zero. The boreal level had too few degrees of freedom ($df_s < 4$), which invalidates the Satterthwaithe approximation (*i.e.*, calculation of the effective $df_s$ of a linear combination of independent sample variances; [63, 65, 66]). Whether it is significant or not, we cannot interpret the boreal result with strong statistical confidence. Thus, the results of the RVE with this moderator in the equation have to be interpreted with caution. On

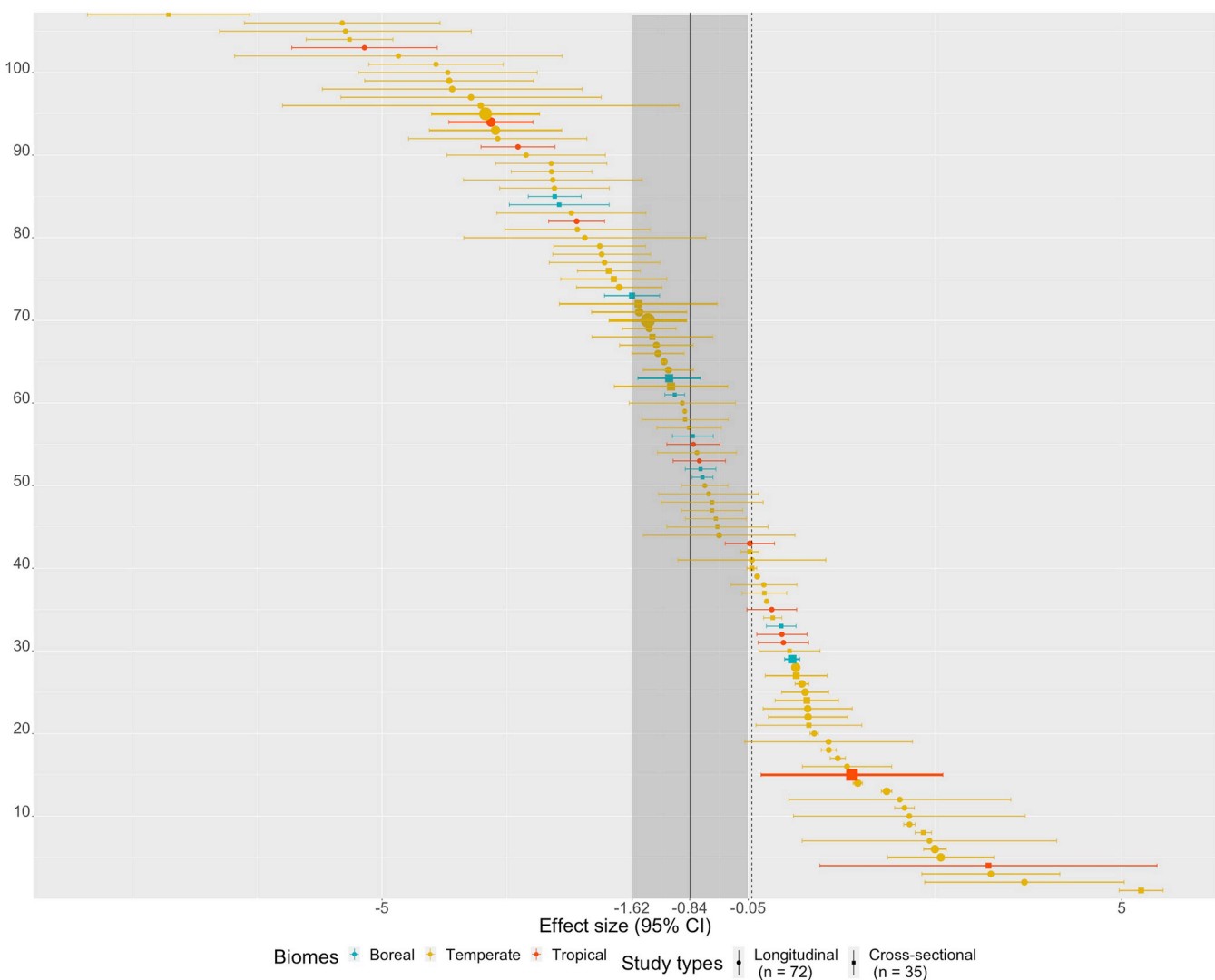

**Fig 3. Forest plot of the meta-analysis.** The mean effect size is -0.84 (95% CI = -1.62 to -0.05, shaded grey area), where study type is shape-coded (i.e., circle for longitudinal studies and squares for cross-sectional studies) and biome color coded (i.e., boreal in blue, temperate in yellow and tropical in red). A negative effect size means that there is a negative impact of hydropower in impacted sites as opposed to reference sites, whereas a positive effect size means that there is positive impact of hydropower in impacted sites as opposed to reference sites.

the contrary, both type of impact and study moderators had enough $df_s$ (for all levels) for robust statistical analysis. Statistically significant effects were neither found for downstream flow regulation (estimate = -0.50; 95% CI = -1.24 to 0.25; P = 0.18) nor for upstream water level fluctuations/drawdown impact types (estimate = -1.20; 95% CI = -3.45 to 1.05; P = 0.26; Fig 4B). When study type was used as a moderator, there was no significant difference in effect sizes for cross-sectional design studies (*i.e.*, natural versus impacted; estimate = 0.45; 95% CI = -1.14 to 2.03; P = 0.56) and longitudinal gradient type studies (*i.e.*, spatial gradient; estimate = -1.11; 95% CI = -2.32 to 0.10; P = 0.07; Fig 4C). However, we can observe a visual trend where studies that were considered as gradients were most often associated with negative effect sizes (not supported statistically). As for results from the season moderator, conclusions can only be drawn for the summer and winter levels. Statistically significant effects were neither found for summer (estimate = -0.46; 95% CI = -2.26 to 1.24; P = 0.58) nor for winter (estimate = -0.39;

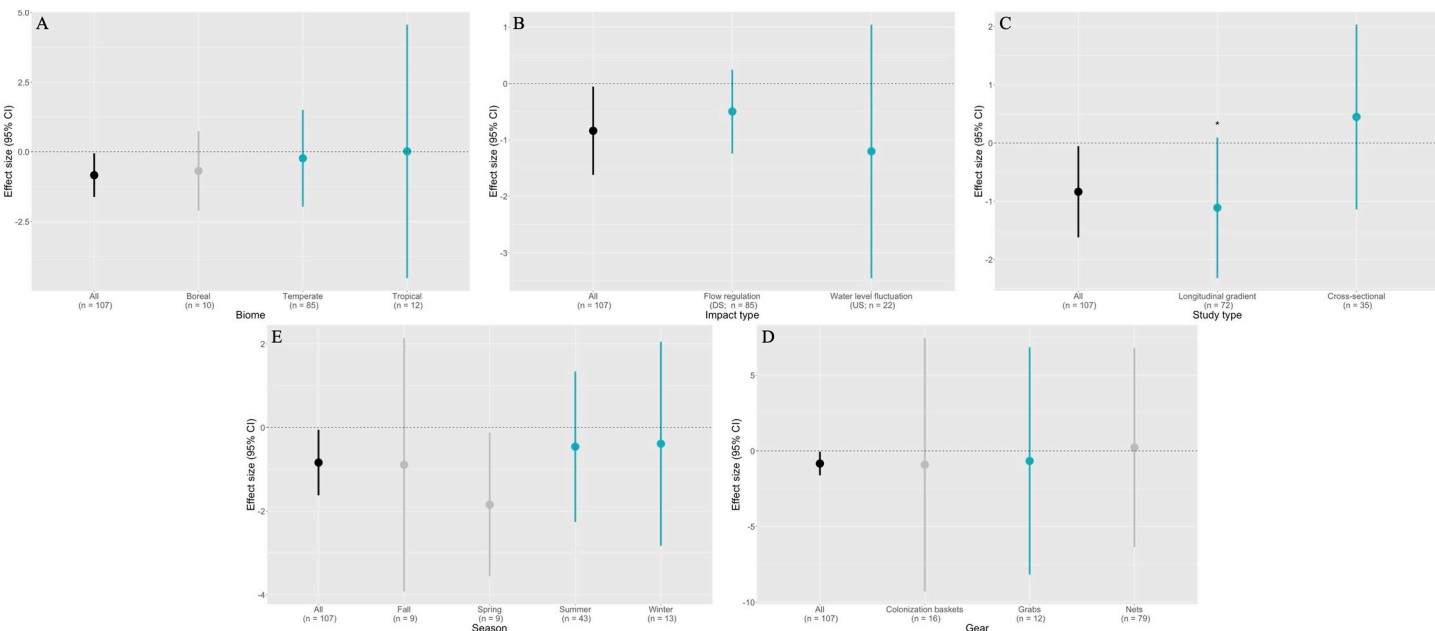

**Fig 4. Plots showing the mean effect sizes and their confidence interval for each of the moderators.** Value in black is the mean effect size of the meta-analysis and the other colors are related to the different effect sizes when including specific moderators. When in grey, statistical significance of moderator cannot be interpreted with confidence due to statistical power issues ($df_s$ insufficient). When effect size is in color (i.e., blue) statistical interpretation can be made with confidence, whether it is significant or not (sufficient $df_s$). Asterisk signifies statistically marginally significant effect.

95% CI = -2.83 to 2.05; P = 0.70; Fig 4D). Results are similar for the sampling gear moderator, statistical inference can only be drawn for the grab level, where no statistically significant effect can be observed (estimate = -0.67; 95% CI = -8.18 to 6.85; P = 0.83; Fig 4E), other levels cannot be interpreted with confidence due to insufficient $df_s$ (Satterthwaithe approximation invalidated).

## Discussion

The meta-analysis conducted on 107 pairs of observations (*i.e.*, impacted versus reference) extracted from 24 studies suggests that hydropower does statistically impact macroinvertebrates richness in a clear directional way. The richness of macroinvertebrates in dams and reservoirs impacted by hydropower is significantly lower than in unimpacted ecosystems (Fig 2). To our knowledge, this meta-analysis is the first to resolve the result divergence observed in the existing literature. In fact, more than 88% of the effect sizes were significantly different from zero with 56% of the observations showing reduced richness due to hydropower and 32% of the observations showing reduced richness in reference conditions. These percentages, along with the statistically significant mean effect size (*i.e.*, -0.84; 95% CI = -1.62 to -0.05; P = 0.04), suggest a predominance of reduced macroinvertebrate richness due to hydropower, dams and reservoirs.

Nevertheless, we also observed a large heterogeneity in macroinvertebrate richness responses to hydropower (Fig 3). Part of this variability across observations and studies can usually be explained by environmental and methodological variability, which we can try to control using moderators. In our meta-analysis, we considered biome (*i.e.*, boreal, temperate or tropical), type of impact (*i.e.*, water level fluctuations in the reservoir or flow regulation downstream of the dam), the type of study (*i.e.*, spatial longitudinal gradient [upstream vs downstream] or cross-sectional [natural lake versus impacted reservoir]), sampling season (*i.*

*e.*, spring, summer, fall and winter) and sampling gear (*i.e.*, net, grab and colonization basket) as moderators. Despite the documented effects on macroinvertebrate richness of the moderators included in our analysis [27, 29, 30, 32, 33], none of our moderators statistically explained heterogeneity in macroinvertebrate richness responses to hydropower.

## Biome moderator

In general, it is quite common to observe a latitudinal gradient in species richness, where there is a maximum richness in the tropics and a decline towards the poles [30, 31, 67, 68]. In a meta-analysis, Turgeon et al. [69], showed that the impacts of impoundments on fish richness was different across biomes. Significant declines in richness were observed in the tropics, a lower decline was observed in temperate regions and no impacts was observed in boreal biomes. Thus, we hypothesized that a latitudinal gradient could also influence the mean effect size in macroinvertebrates (*e.g.*, losing one species over a few species is more costly in terms of biodiversity loss, than over multiple species). Our data did not support a latitudinal trend which is not too surprising because there is no clear pattern about whether or not macroinvertebrate richness follows a latitudinal gradient [70–72]. Pearson and Boyero [73] observed a richness peak around the equator for dragonflies (*i.e.*, odonata), but no clear global pattern for caddisflies (*i.e.*, trichoptera), whereas Vinson and Hawkins [71] observed richness peaks at mid-latitudes in South and North America for mayflies (*i.e.*, ephemeroptera), stoneflies (*i.e.*, plecoptera) and caddisflies (*i.e.*, trichoptera; [EPT] orders). On the contrary, Scott et al. [74] did not observe this latitudinal gradient for EPT in northern Canada. Geographic range (*i.e.*, narrow range and exclusion of extreme latitudes; [71]), sampling effort [75], macroinvertebrate specific life-history strategies [73, 74] and data resolution (*i.e.*, weak gradient for local species richness; [76]) were probably the reason behind this lack of consensus [77], and absence of a trend in this meta-analysis. Moreover, the unbalanced sample size across biomes may have impeded any clear conclusions. Boreal samples represented 9% of the data, tropical samples 11% and roughly 79% of the data belong to temperate ecosystems. This prevented us from statistically supporting a trend for both boreal and tropical observations.

## Impact type moderator

Water level fluctuation in reservoirs is characterized by yearly drawdown (*i.e.*, long-term oscillations in the water level), whereas flow regulation usually is reflected by daily or weekly changes in flow downstream of a dam (*i.e.*, short-term oscillations; [20]), keeping in mind that variation in the reservoir are also reflected downstream. In reservoir experiencing water level fluctuations in the form of winter drawdown (21% of the reservoirs in this meta-analysis; littoral zone samples), the littoral zone is exposed to desiccation and freezing for an extended period of time, which caused a loss of macroinvertebrate taxa [20, 21, 78, 79] and decreased their overall abundance [80]. In the case of daily downstream fluctuations (79% of the reservoirs in this meta-analysis), organisms can burrow in the sediment (*i.e.*, low mobility taxa such as oligochaetes) or follow the water level up and down (*i.e.*, high mobility swimming taxa such as dragonfly nymphs; [20]). Riverine organisms have evolved and developed adaptations to survive flood and drought hydrological dynamics [81], but lentic taxa are not adapted to extreme water level fluctuations, such as winter drawdown in reservoirs ($> 2m$ amplitude; [22]). Because of that, we were expecting a higher impact and effect size of water level fluctuation in reservoir (*i.e.*, winter drawdown) on macroinvertebrates richness than downstream flow regulation but found no difference nor significant effect of impact type moderator.

## Study type moderator

Another moderator that could explain heterogeneity in our results was the type of study. An ideal way to analyze the impact of hydropower on richness is to set the reference conditions as richness before impoundment versus impacted conditions as richness after impoundment, within the same ecosystem (*e.g.*, a reference river that was impounded into an impacted reservoir; longitudinal in time, also known as BACI; [32]). However, studies using this type of methodology are rare, even more so for macroinvertebrates, and such studies were mostly absent in the results from the literature review and if present, did not fulfill the required criteria to be included in the meta-analysis. Here, we accounted for two types of study methodologies; a cross-sectional methodology (*i.e.*, comparing a hydropower impacted ecosystem with a natural/reference ecosystem; 33%) and longitudinal gradient in space methodology (*i.e.*, upstream of a dam versus downstream of that same dam, within a single ecosystem; 67%), which are both considered as simple CI studies [32]. These methodologies are not ideal, compared to the longitudinal in time methodology (*i.e.*, BACI or BA studies), as their reference spatial point differs from the impacted spatial point, thus introducing some environmental noise and there is no way to control for environmental stochasticity [32]. In the cross-sectional type, studies included inter-ecosystem's variation (*i.e.*, impacted ecosystem was spatially independent from reference/non-impacted ecosystem). This inherently added some heterogeneity in the responses and the impacts could be more difficult to detect. In the longitudinal in space studies (reference upstream and impacted downstream of the dam, at a single point in time), the observations were from the same ecosystem. Here, there is less heterogeneity and thus, we could have expected the results to be less variable. There was no significant difference between the two study types in this meta-analysis. Even though the patterns were not significant, we observed a visual negative trend in the spatially longitudinal gradient studies (*i.e.*, higher richness in reference ecosystems, that is upstream of the dam), with a slightly tighter range of variation (not statistically supported). The cross-sectional studies had a tendency toward higher richness in reservoirs, with a larger range of variation. This might highlight a problem with general study design and the choice of the reference ecosystem, and caution is in order when interpreting these trends. Nevertheless, we believe that cross-sectional and longitudinal in space references are the best benchmark available to overcome current limitations regarding the lack of richness data before impoundment in the literature.

## Sampling season and gear moderators

We, initially thought we would observe an effect of sampling season when comparing effect sizes since diversity and abundance is known to fluctuate yearly [33] but there was no statistically significant effect of any season that could modulate the meta-analysis outcome. Similarly, we though that depending on the sampling gear we would observe varying diversity because different stratum would be sampled, thus representing different communities of macroinvertebrates [27] but no additional amount of heterogeneity was explained by the sampling gear moderator. The unbalanced nature of the sample sizes could be one of the reasons why these moderators do not provide statistically significant inferences.

The analysis of moderators did not allow a better understanding of the large heterogeneity in the effect sizes and suggests that maybe other moderators, which were not available for the studies included in this meta-analysis, could help tease out some of that heterogeneity. For instance, given that macroinvertebrate are very adapted to their localized environmental conditions (*e.g.*, temperature, granulometry, wave disturbance and macrophytes; [34]), their richness maybe regulated at a much finer scale. Thus, these micro-habitat moderators could be

especially relevant to include in a future meta-analysis, although very hard to collect in such a global consolidating endeavour.

## Conclusion

This meta-analysis suggested that there is a clear, directional, statistically significant conclusion regarding whether or not hydropower impacts macroinvertebrate richness; macroinvertebrate richness in hydropower, dams and reservoirs impacted environments is significantly lower than in unimpacted environments. However, we also observed a large range of effect sizes, from very negative to very positive impacts of hydropower. The environmental and methodological heterogeneity in the studies might have hindered the detection of a stronger significant effect, unfortunately none of our moderators helped untangle that heterogeneity. This advises that other moderators, not included in this study due to unavailability among the studies, may be responsible for some of that heterogeneity. We advocate that local, smaller-scale variables pertaining to habitat physicochemical characteristics may bring some clarity about the large heterogeneity in effect sizes. As new studies evaluating the impacts of hydropower on macroinvertebrate richness accumulate, we would recommend that information regarding local habitat variables be available so they could be recorded and evaluated as moderators in future meta-analyses. This meta-analysis was able to highlight a clear directional effect of hydropower on macroinvertebrate richness. As richness is only one aspect of biodiversity, it would be interesting, in future studies, to conduct additional analyses of community composition and functional diversity and thus, get a better portrait of the impact of hydropower on macroinvertebrate biodiversity, not only richness. Moreover, as new studies are available, it would be interesting to keep enriching this meta-analysis to see if the results statistical confidence could strengthen and become even more assertive.

## Supporting information

**S1 Fig. Funnel plot for this meta-analysis.** No statistically significant asymmetry is observed (z = -1.74, P = 0.08).
(DOCX)

**S1 Table. PRISMA checklist.** Specific to this meta-analysis, from Moher et al. *[37]*.
(DOCX)

**S2 Table. Metadata table showing all variables for each study included in this meta-analysis.** Use S3 Table as a companion table to get more information on each variable.
(DOCX)

**S3 Table. Companion table describing all variables in S2 Table.**
(DOCX)

**S4 Table. Table showing the sensitive analysis outputs.** Rho values ($\rho$) ranges from 0 to 1 and mean effect size (ES), standard error (SE) and between study variance ($\tau^2$) estimates are relatively insensitive to these varying $\rho$ values.
(DOCX)

## Acknowledgments

We thank the CIRAIG–Polytechnique Montréal for covering the publication fees. We also thank the CSBQ for offering systematic reviews and meta-analyses workshops, which were more than useful for putting together this meta-analysis research article.

## Author Contributions

**Conceptualization:** Gabrielle Trottier, Daniel Boisclair, Cécile Bulle, Manuele Margni.

**Data curation:** Gabrielle Trottier.

**Formal analysis:** Gabrielle Trottier.

**Funding acquisition:** Gabrielle Trottier, Manuele Margni.

**Investigation:** Daniel Boisclair, Cécile Bulle, Manuele Margni.

**Methodology:** Gabrielle Trottier.

**Project administration:** Gabrielle Trottier, Manuele Margni.

**Supervision:** Katrine Turgeon, Daniel Boisclair, Cécile Bulle, Manuele Margni.

**Validation:** Katrine Turgeon.

**Visualization:** Gabrielle Trottier.

**Writing – original draft:** Gabrielle Trottier.

**Writing – review & editing:** Gabrielle Trottier, Katrine Turgeon, Daniel Boisclair, Cécile Bulle, Manuele Margni.

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
