## [Decision Letter · Decision Letter 0]

25 Jun 2022

PONE-D-22-07760The impacts of hydropower on freshwater macroinvertebrate richness: A global meta-analysisPLOS ONE

Dear Dr.Trottier

Thank you for submitting your manuscript to PLOS ONE. After careful consideration, we think that it has merit but but it requires a minor revision. Therefore, we invite you to submit a revised version of the manuscript that addresses the points raised during the review process.

 In particular, please give additional rationale why you focused specifically on richness, instead of biodiversity and community assemblage. Also explain and justify your rationale for using benthic grabs instead of nets. There is a discrepancy in the number of included papers (24) and the 17 studies from which you collected metrics. Please see the reviewer's comments.

We look forward to receiving your revised manuscript.

Kind regards,

Jay Richard Stauffer, Jr.

Academic Editor

PLOS ONE

Journal Requirements:

Reviewers' comments:

Reviewer's Responses to Questions

**Comments to the Author**

1. Is the manuscript technically sound, and do the data support the conclusions?

Reviewer #1: Yes

Reviewer #2: Yes

2. Has the statistical analysis been performed appropriately and rigorously? 

Reviewer #1: Yes

Reviewer #2: Yes

3. Have the authors made all data underlying the findings in their manuscript fully available?

Reviewer #1: Yes

Reviewer #2: Yes

4. Is the manuscript presented in an intelligible fashion and written in standard English?

Reviewer #1: Yes

Reviewer #2: Yes

5. Review Comments to the Author

Reviewer #1: Review of Trottier et al.

The authors of this manuscript, “The impacts of hydropower on freshwater macroinvertebrate richness: a global meta-analysis” assess the impacts of hydropower dams on macroinvertebrate richness. They conduct a meta-analysis of the literature, and attempt to account for several potential moderators, including the biome the studies occurred in; whether the study sites occurred upstream (within the reservoir) or downstream of the dam; the study design; the sampling season; and the type of sampling gear used. Using a random and mixed effect modelling approach, the authors conclude that hydropower does significantly impact macroinvertebrate richess, but they observed a large range of effect sizes that were not accounted for by the included moderators, and suggest including local habitat variables in future studies would be helpful in understanding the response of macroinvertebrate richness to hydropower impacts.

Overall, I thought the meta-analysis was well laid out and clear (following the PRISMA approach), however, there were a few areas of the manuscript that were confusing and require clarification. The authors focus their meta-analysis specifically on macroinvertebrate richness. In their introduction, they mention three aspects of biodiversity: richness, community assemblage and functional diversity. It might be helpful to provide some additional rationale as to why the decision to focus specifically on richness. Is richness more easily quantifiable and extractable for a metaanalyses? While functional diversity measures are likely more limited in the literature, community composition data is frequently assessed, and might also be insightful in looking at hydropower impacts. This might be beyond the scope of this paper, but perhaps some additional rationale to focusing on richness may be helpful.

The authors included sampling gear as a proxy for habitat stratum (i.e. benthic grabs vs nets). This approach might need a caveat, as many biomonitoring protocols (including the EPA in the US) use nets for multiple-habitat assessments, whereas grabs are primarily used in deeper reservoir studies (upstream).

The authors included a comprehensive literature search using the Web of Science core collection, and provide the terms used. In the research strategy (line 180; Figure1), the authors state that they reduced the number of included papers to 24, but in the data collection section (Line 192) mention collecting metrics from 17 studies. There does not seem to be any mention on how the number of studies was reduced from 24 (Table S2 does list 24 studies).

The seasonal sampling moderator (spring/summer/fall/winter) seems very coarse. For either upstream draw down or downstream modifications in stream flow, were samples collected immediately after alterations or later in the season? As some macroinvertebrates are relatively mobile, they could recolonize shortly after short-term water regulation effects.

In the data collection/extraction section, it might be helpful to include information on whether multiple authors were involved in abstracting the data, or if this was done by a single analyst.

For samples from reservoirs, it might be useful to know whether samples were collected in the center of the lake or in the littoral regions. The authors state (Line 415) they were expecting a large effect of water level fluctuations. If these were samples collected from the profundal region in the center of the lake, water level drawdown might not have a large impact.

Minor edits:

Line 373: The authors state “despite the documented effects on macroinvertebrate richness..” while these are discussed briefly elsewhere in the manuscript, It would be helpful to include some citations with this statement.

Line 399: states that the authors did not have a statistical supporting trend for both boreal and tropical, but previously (Lines 319-321), they state they did make statistically valid inferences for both temperate and tropical biomes (just not for boreal).

Overall, I thought the meta-analysis was interesting, Perhaps it might be worth mentioning that in the discussing/conclusion that additional analyses of community composition and functional diversity would also be an important future direction in assessing the impacts of hydropower on macroinvertebrate communities. As mentioned by the authors in their introduction, taxon richness is only one aspect of biodiversity.

Reviewer #2: This paper describes a meta-analysis of the literature asking whether hydropower, dams, and reservoirs affect macroinvertebrate species richness. The authors found that a preponderance of studies found negative effects on species richness. However, there was a range of effect sizes in their dataset and despite having a number of likely moderators that might explain variation between studies, the authors found that these moderators were not effective in explaining effect size variation.

Generally this study was well conceived and completed. I liked the detailed description of how the meta-analysis was done and that they used the PRISMA methodology for conducing a metaanalysis.

One thing I’d like to see clarified is what is being asked of the moderators. Are you asking if the moderators impact species richness, or if the moderators explain heterogeneity in the effect sizes? I think it is the latter, but this could be made more clear. I suppose that’s why you call them moderators and not covariates?

Although I felt there was a lack of detail explaining this aspect of the study, there are at times places where it felt like there was too much detail (see specific comments below).

In addition, a careful screening of the paper would be useful to address various grammatical mistakes – I’ve highlighted some below – which will make the paper more easily assessable to the reader.

Minor comments:

Line 29 and throughout (e.g., 258, 264, 267, 268): mixed effects models are mixed because they contain fixed and random effects. It is therefore redundant to say “random and mixed effect model”.

Line 32: change “statistical way” to “statistically significant way”.

Line 52: add comma before “but”

Line 69: between <what> and “natural-like” flows?

Line 71: delete “allow to”

Line 81: Why would this latitudinal effect on species richness affect effect sizes? Perhaps more likely to see effects in tropics than higher latitudes because a floor effect makes it difficult to detect change in richness? In any case, it would help the reader to explain the reasoning rather than leaving the reader to connect the dots.

Lines 88-97: Most of these study designs are not in your dataset – or even likely as you point out. What isn’t mentioned here are the two study designs you did use in your models.

Lines 121-138: A little overly wordy – does meta-analysis really need this much defense as a method for the audience of PLoS One?

Line 154: Why this date (1989)?

Line 168: Why was sorting required? If you went through every result, then sorting is irrelevant.

Line 170: Change “studies” to “study”

Line 192: Everywhere else in the manuscript you say “24” studies.

Lines 241-243: This belongs in the next section, not here.

Lines 291-292: You call a “P = 0.07” “marginally significant” below.

Lines 298-299: How is the “true heterogeneity” compared to the “methodological variability” assessed? Is this left-over variance not explained by your model?

Line 300: Change “will be” to “was”.

Lines 302-304: You mean hydropower, dams, and reservoirs DID have a statistically significant effect on macroinvertebrate richness, right?

Line 322: Change “little” to “few”.

Line 326: add “for all levels” after “df”

Lines 353-354: Sentence begins in a way redundant with previous sentence. Change “Hydropower impacts macroinvertebrates, their richness…” to “Macroinvertebrate richness…”

Line 358: Change “from which” to “with”

Lines 364-365: Why are you now calling this a marginal effect?

Lines 367-368: Delete “and thus potentially strengthen the statistical significance of the result”. Awkward phrasing that doesn’t seem necessary.

Line 407: Change “been causing” to “caused”.

Lines 426-427: Delete “(none could be included in this case)”. Redundant.</what>

6. PLOS authors have the option to publish the peer review history of their article (what does this mean?). If published, this will include your full peer review and any attached files.

Reviewer #1: No

Reviewer #2: No

---

## [Author Response · Author response to Decision Letter 0]

28 Jul 2022

Reviewers' comments:

Reviewer's Responses to Questions

Comments to the Author

1. Is the manuscript technically sound, and do the data support the conclusions?

Reviewer #1: Yes

Reviewer #2: Yes

2. Has the statistical analysis been performed appropriately and rigorously?

Reviewer #1: Yes

Reviewer #2: Yes

3. Have the authors made all data underlying the findings in their manuscript fully available?

Reviewer #1: Yes

Reviewer #2: Yes

4. Is the manuscript presented in an intelligible fashion and written in standard English?

Reviewer #1: Yes

Reviewer #2: Yes

5. Review Comments to the Author

REVIEWER #1

Review of Trottier et al.

The authors of this manuscript, “The impacts of hydropower on freshwater macroinvertebrate richness: a global meta-analysis” assess the impacts of hydropower dams on macroinvertebrate richness. They conduct a meta-analysis of the literature, and attempt to account for several potential moderators, including the biome the studies occurred in; whether the study sites occurred upstream (within the reservoir) or downstream of the dam; the study design; the sampling season; and the type of sampling gear used. Using a random and mixed effect modelling approach, the authors conclude that hydropower does significantly impact macroinvertebrate richness, but they observed a large range of effect sizes that were not accounted for by the included moderators, and suggest including local habitat variables in future studies would be helpful in understanding the response of macroinvertebrate richness to hydropower impacts.

Overall, I thought the meta-analysis was well laid out and clear (following the PRISMA approach), however, there were a few areas of the manuscript that were confusing and require clarification. The authors focus their meta-analysis specifically on macroinvertebrate richness. In their introduction, they mention three aspects of biodiversity: richness, community assemblage and functional diversity. It might be helpful to provide some additional rationale as to why the decision to focus specifically on richness. Is richness more easily quantifiable and extractable for a meta-analyses? While functional diversity measures are likely more limited in the literature, community composition data is frequently assessed, and might also be insightful in looking at hydropower impacts. This might be beyond the scope of this paper, but perhaps some additional rationale to focusing on richness may be helpful.

*** Response: As it was pointed out by the reviewer, we focused this meta-analysis on macroinvertebrate richness because it was more easily quantifiable/extractable in this context. It took a lot of effort to access simple taxon richness data for 24 studies. As community composition and functional diversity are more data hungry and a lot more limited in term of accessibility and computability than taxon richness, they were not as easy to retrieve at the meta-analysis’ scale. We added a statement at the end of the introduction (lines 134-137), as well as a sentence saying it would be interesting to conduct further studies using these biodiversity aspects in the conclusion (lines 601-604).

The authors included sampling gear as a proxy for habitat stratum (i.e. benthic grabs vs nets). This approach might need a caveat, as many biomonitoring protocols (including the EPA in the US) use nets for multiple-habitat assessments, whereas grabs are primarily used in deeper reservoir studies (upstream).

*** Response: We thank the reviewer for bringing this issue to our attention. We added a caveat inspired by the reviewer’s suggestion (lines 115-119).

The authors included a comprehensive literature search using the Web of Science core collection, and provide the terms used. In the research strategy (line 208; Figure1), the authors state that they reduced the number of included papers to 24, but in the data collection section (Line 220) mention collecting metrics from 17 studies. There does not seem to be any mention on how the number of studies was reduced from 24 (Table S2 does list 24 studies).

*** Response: This was a typographical error. It was corrected to 24 studies. 

The seasonal sampling moderator (spring/summer/fall/winter) seems very coarse. For either upstream draw down or downstream modifications in stream flow, were samples collected immediately after alterations or later in the season? As some macroinvertebrates are relatively mobile, they could recolonize shortly after short-term water regulation effects.

*** Response: This is true. However, since this is a meta-analysis, it was not easy to choose an appropriate coarseness for this specific moderator, especially considering some of the studies were not very specific about their sampling. We could have used Julian day, but we were more interested in the coarser temporal scale effect of seasons rather than micro changes in colonization following immediate water regulation. We have added justification to this choice in the manuscript (lines 131-133).

In the data collection/extraction section, it might be helpful to include information on whether multiple authors were involved in abstracting the data, or if this was done by a single analyst.

*** Response: We added this information to the data collection paragraph (lines 220-221). 

For samples from reservoirs, it might be useful to know whether samples were collected in the center of the lake or in the littoral regions. The authors state (Line 518) they were expecting a large effect of water level fluctuations. If these were samples collected from the profundal region in the center of the lake, water level drawdown might not have a large impact.

*** Response: After verification, all samples from reservoir studies were collected in the littoral zone, not in the profundal zone. We specified types of samples at line 518. 

Minor edits:

Line 477: The authors state “despite the documented effects on macroinvertebrate richness.” while these are discussed briefly elsewhere in the manuscript, It would be helpful to include some citations with this statement.

*** Response: We added appropriate references for this statement. 

Line 496: states that the authors did not have a statistical supporting trend for both boreal and tropical, but previously (lines 395-397), they state they did make statistically valid inferences for both temperate and tropical biomes (just not for boreal).

*** Response: We could indeed make valid statistical inferences, but for both temperate and tropical biome, the confidence interval encompassed zero, so we can say with confidence that there is no statistically significant effect of tropical and temperate biomes, whereas for the boreal biome, we cannot even interpret with confidence the statistical trend observed. We added clarifications to minimize reader confusion (lines 395-398)

Overall, I thought the meta-analysis was interesting, perhaps it might be worth mentioning that in the discussing/conclusion that additional analyses of community composition and functional diversity would also be an important future direction in assessing the impacts of hydropower on macroinvertebrate communities. As mentioned by the authors in their introduction, taxon richness is only one aspect of biodiversity.

*** Response: We thank the reviewer for its valuable feedback. We added a sentence mentioning the future relevance of additional community composition and functional diversity analyses (lines 601-604).

REVIEWER #2

This paper describes a meta-analysis of the literature asking whether hydropower, dams, and reservoirs affect macroinvertebrate species richness. The authors found that a preponderance of studies found negative effects on species richness. However, there was a range of effect sizes in their dataset and despite having a number of likely moderators that might explain variation between studies, the authors found that these moderators were not effective in explaining effect size variation.

Generally this study was well conceived and completed. I liked the detailed description of how the meta-analysis was done and that they used the PRISMA methodology for conducing a meta-analysis.

One thing I’d like to see clarified is what is being asked of the moderators. Are you asking if the moderators impact species richness, or if the moderators explain heterogeneity in the effect sizes? I think it is the latter, but this could be made more clear. I suppose that’s why you call them moderators and not covariates?

*** Response: We are indeed asking the moderators to explain additional heterogeneity, as suggested by the reviewer. We added a sentence to clarify this matter in the manuscript (lines 312-314).

Although I felt there was a lack of detail explaining this aspect of the study, there are at times places where it felt like there was too much detail (see specific comments below).

In addition, a careful screening of the paper would be useful to address various grammatical mistakes – I’ve highlighted some below – which will make the paper more easily assessable to the reader.

Minor comments:

Line 30 and throughout (e.g., 292, 302, 303, 305): mixed effects models are mixed because they contain fixed and random effects. It is therefore redundant to say “random and mixed effect model”.

*** Response: We thank the reviewer for bringing this matter to our attention. We reviewed this formulation and felt comfortable using “random/mixed-effects model” (as it is used in Viechtbauer [2010]) instead of “random and mixed effect model” throughout the manuscript. 

Line 33-34: change “statistical way” to “statistically significant way”.

*** Response: We changed it according to the reviewer suggestion.

Line 55: add comma before “but”

*** Response: We added a comma.

Line 72: between and “natural-like” flows?

*** Response: We added “impacted flows” (it became missing during the back and forth between co-authors revision of the manuscript pre-submission). 

Line 74: delete “allow to”

*** Response: We thank the reviewer for taking the time to highlight these grammatical/syntax errors, we deleted “allow to”.

Line 83-84: Why would this latitudinal effect on species richness affect effect sizes? Perhaps more likely to see effects in tropics than higher latitudes because a floor effect makes it difficult to detect change in richness? In any case, it would help the reader to explain the reasoning rather than leaving the reader to connect the dots.

*** Response: We are not entirely sure what the reviewer meant by “floor effect”. Nonetheless, we were able to clarify our reasoning and added few lines at two locations in the manuscript (lines 85-87 and 488-496) stating that the loss of one taxon over few taxa is relatively more important than over multiple taxa and thus, why we could potentially observe an effect of a latitudinal gradient on our effect size (i.e., species richness highest in the tropics and lowest near the poles).

Lines 99-104: Most of these study designs are not in your dataset – or even likely as you point out. What isn’t mentioned here are the two study designs you did use in your models.

*** Response: The study designs found in our meta-analysis are in facts two variants of the Control-Impact sampling design (line 128-131), that is cross-sectional (i.e., reference natural lake versus impacted reservoir) and longitudinal spatial gradient (i.e., upstream of a dam [reservoir] versus downstream of a dam [river]). In lines 99-104, we simply make our case that sampling, or study design could affect the conclusions of a study. 

Lines 137: A little overly wordy – does meta-analysis really need this much defense as a method for the audience of PLoS One?

*** Response: We deleted this paragraph. We agree meta-analysis does not need as much defense and description in PLOS ONE paper.

Line 183: Why this date (1989)?

*** Response: This is the earliest year you can search for articles in Web of Science, we specified it in the manuscript.

Line 198: Why was sorting required? If you went through every result, then sorting is irrelevant.

*** Response: We thank the reviewer for pointing that out. We deleted this part of the sentence.

Line 198: Change “studies” to “study”

*** Response: We changed “studies” to “study”.

Line 220: Everywhere else in the manuscript you say “24” studies.

*** Response: This was a typographical error. It has been corrected to 24.

Lines 287-290: This belongs in the next section, not here.

*** Response: We thank the reviewer for its feedback on the matter. However, without further justifications as to why this belong in the next section (next paragraph or in the results?), we feel confident that it does belong where it is now and would prefer that it remains that way. 

Lines 357-359: You call a “P = 0.07” “marginally significant” below.

*** Response: We rectified our statement regarding the p-value of 0.07 stating it was not statistically significant, but that visual assessment could detect a potential trend where studies that were considered as gradients were most often associated with negative effect sizes (not supported statistically; lines 408-413).

Lines 362-365: How is the “true heterogeneity” compared to the “methodological variability” assessed? Is this left-over variance not explained by your model?

*** Response: The methodological variability is the heterogeneity due to sampling error within each individual study included in the meta-analysis, whereas the true heterogeneity is due to specific study characteristics, which can introduce heterogeneity among the effect sizes. The combination of this methodological and true heterogeneity is referred to as total heterogeneity. We tried to clear that up in the methodology section (lines 279-290) so that lines 362-365 are easier to understand. 

Line 366: Change “will be” to “was”.

*** Response: We changed “will be” to “was”.

Lines 369: You mean hydropower, dams, and reservoirs DID have a statistically significant effect on macroinvertebrate richness, right?

*** Response: Yes, this has been rectified.

Line 399: Change “little” to “few”.

*** Response: We changed “will be” to “was”.

Line 404: add “for all levels” after “df”

*** Response: We added “(for all levels)” after “df”.

Lines 448-450: Sentence begins in a way redundant with previous sentence. Change “Hydropower impacts macroinvertebrates, their richness…” to “Macroinvertebrate richness…”

*** Response: This has been rectified.

Line 452: Change “from which” to “with”

*** Response: We modified “from which” to “with” as well as “showed” to “showing” in the remainder of the sentence. 

Lines 467: Why are you now calling this a marginal effect?

*** Response: This is a leftover typographical error from a previous version of this manuscript, we thank the reviewer for pointing it out. We deleted part of the sentence mentioning the “marginality” of the mean effect size which is not valid for the updated/enhanced meta-analysis (17 to 24 studies included). 

Lines 472: Delete “and thus potentially strengthen the statistical significance of the result”. Awkward phrasing that doesn’t seem necessary.

*** Response: We thank the reviewer for its output, we deleted this part of the sentence. 

Line 516: Change “been causing” to “caused”.

*** Response: We changed “been causing” to “caused”.

Lines 539: Delete “(none could be included in this case)”. Redundant.

*** Response: We deleted this parenthesis. 

6. PLOS authors have the option to publish the peer review history of their article (what does this mean?). If published, this will include your full peer review and any attached files.

Do you want your identity to be public for this peer review? For information about this choice, including consent withdrawal, please see our Privacy Policy.

Reviewer #1: No

Reviewer #2: No

---

## [Editor Report · Decision Letter 1]

3 Aug 2022

The impacts of hydropower on freshwater macroinvertebrate richness: A global meta-analysis

PONE-D-22-07760R1

Dear Dr. Trottier

We’re pleased to inform you that your manuscript has been judged scientifically suitable for publication and will be formally accepted for publication once it meets all outstanding technical requirements.

Kind regards,

Jay Richard Stauffer, Jr.

Academic Editor

PLOS ONE

Additional Editor Comments (optional):

I have made a few minor suggestions in the attached manuscript. The paper is now accepted for publication
---

## [Editor Report · Acceptance letter]

8 Aug 2022

PONE-D-22-07760R1 

The impacts of hydropower on freshwater macroinvertebrate richness: A global meta-analysis 

Dear Dr. Trottier:

I'm pleased to inform you that your manuscript has been deemed suitable for publication in PLOS ONE. Congratulations! Your manuscript is now with our production department. 

Kind regards, 

on behalf of

Dr. Jay Richard Stauffer, Jr. 

Academic Editor

PLOS ONE